# GNSS and Photogrammetric UAV Derived Data for Coastal Monitoring: A Case of Study in Emilia-Romagna, Italy

Enrica Vecchi [1,*], Luca Tavasci [1], Nunzio De Nigris [2] and Stefano Gandolfi [1]

1 Department of Civil, Chemical, Environmental and Materials Engineering (DICAM), University of Bologna, 40136 Bologna, Italy; luca.tavasci@unibo.it (L.T.); stefano.gandolfi@unibo.it (S.G.)
2 Agency for the Prevention, Environment, and Energy of Emilia-Romagna (Arpae), 40136 Bologna, Italy; ndenigris@arpae.it
* Correspondence: enrica.vecchi@unibo.it

**Abstract:** GNSS real-time techniques and UAV photogrammetry can be alternative methods for the monitoring of sand beaches. This activity is particularly important in environments such as the Emilia-Romagna coastline. In this paper, two couples of surveys (year 2019 and 2020) performed using GNSS or a low-cost UAV equipment over a common area were compared in order to analyse: point-wise height differences, profile shapes along defined sections, and volumes variations over time. Both surveys were aligned to the same reference benchmark through GNSS measurements. The highest discrepancies between the two surveying methods (tens of cm) were found in vegetated areas and along the shoreline, otherwise, the height differences are mainly within the 10 cm level. In terms of volumes, excluding the most critical areas, differences close to zero can be found. Obtained results show that GNSS and UAV photogrammetry provides similar results, at least for quite flat terrains and when decimetre-level accuracy is required.

**Keywords:** UAV photogrammetry; GNSS kinematic survey; RTK; Adriatic coast; RGC; sand-beach monitoring





## 1. Introduction

The sharp increase in seaside tourism and the consequent development of Emilia-Romagna coastal areas over the years, have intensified anthropic pressure on beaches, due to the realization of many facilities [1,2]. Nowadays, public authorities are aware of the need to protect beaches from erosion to preserve their environmental relevance and long-term attraction as tourist destinations [2,3], and to therefore avoid environmental and economic losses.

The littoral of Emilia-Romagna in Italy is a significant example of a low sandy coast, characterized by a series of erosion problems which have been worsened by its long-standing urbanization [2,4–6]. For these reasons, research on the possible strategies for coastal preservation have been carried out over the years. The Emilia-Romagna Region have performed several coastal defence interventions to counteract the erosion problem, consisting of both hard structures and soft interventions such as beach nourishments [1,4,7]. In such cases, the monitoring of beach areas is a fundamental tool to obtain a deep knowledge about the state of each coastal stretch in terms of morphological changes, coastline evolution trends, and sediment exchange occurrences. These variations, indeed, could also be induced by the presence of hard structures and by the realization of beach nourishments [7]. Monitoring surveys are therefore necessary to properly design and assess the effects of any intervention. [1,2,8–11]. This awareness led the Emilia-Romagna region to the realization of an intensive monitoring program, consisting of regular surveys of the whole coast every six years, detailed surveys on selected areas for specific purposes, and additional surveys after nourishment interventions.

The effectiveness of monitoring plans strongly depends on the surveying method accuracy, the proper alignment of different datasets, and on the data elaboration phases [8,9,11]. Since there are several possible approaches available for coastal monitoring, each with its specific strengths and weaknesses, it is not possible to identify a single surveying technique for the broad range of coastal applications [9,10,12]. The main discriminating parameter is the spatial resolution necessary to observe the investigated phenomenon and the frequency required to understand the processes and their evolution [8].

Existing geomatic techniques used for coastal applications can be divided into those focused on measuring a limited number of feature points or techniques acquiring massive data (point clouds). The first includes, above all, measurements performed by directly using Global Navigation Satellite Systems (GNSS), while photogrammetry by unmanned aerial vehicles (UAVs) and LIDAR techniques are the most common methods to obtain point clouds in this surveying context.

GNSS surveys are particularly suitable for the monitoring of coastal areas thanks to the absence of obstacles that could reduce sky visibility [7]. The most widespread approaches used in GNSS coastal surveys are those enabling real-time results: Real-Time Kinematic (RTK) and Network-RTK (NRTK) [8,13–16]. The accuracies associated with these methods are a few centimetres. Working in real-time permits the evaluation of the survey quality in the field by checking the state of ambiguity fixing. Moreover, these approaches allow to directly obtain points coordinates during the survey, thus avoiding a post-processing phase. Nevertheless, GNSS surveys become time-consuming if performed on wide areas since they require physical contact of the instrument with all the points to be measured [11]. The spatial density of measured points can be chosen depending on the specific context, nonetheless, interpolation errors are strongly related to the spatial resolution of the acquired dataset. Especially where the area is predominantly flat, such as for sandy beaches, widely spaced tracks can be sufficient to represent the shape of the whole area [17]. For this reason, coastal surveys typically involve a limited number of points acquired following defined transects, i.e., cross and alongshore sections to the coast. GNSS sparse data are to be interpolated to obtain Digital Terrain Models (DTMs), under the assumptions of continuity between different sections [8].

UAV photogrammetry has become common in geomatic applications for environmental monitoring [1,12–14,16,18–21] and many studies have also proved its effectiveness for coastal applications [8,9,15,22,23]. Compared to ground surveys, this technique dramatically reduces human effort in the field, allowing for the acquirement of high-resolution data in a relatively limited time. Nevertheless, this is not totally true, as a set of Ground Control Points (GCPs) distributed over the area is usually required to realize georeferenced models and to avoid distortions. These targets are commonly surveyed using GNSS receivers, thus leading to a significant increase in the time required for the survey. More recently, UAVs equipped with an on-board GNSS receiver capable of acquiring phase observables have been developed to overcome this critical issue. These GNSS receivers can exploit different positioning methods, such as RTK, NRTK, or Post-Processed Kinematic (PPK), enabling accurate positioning of each frame. This reduces the number of necessary GCPs, compared to the common UAV surveys, saving time for ground measurements [24]. Whatever the kind of UAV equipment, the image acquisition is followed by several processing steps to obtain the photogrammetric model. The classical digital photogrammetry is nowadays replaced by the Structure from Motion (SfM) based approach [25–28], which is able to extract 3D information from uncalibrated aerial images. Point clouds are then generated as intermediary products of the process and need to be georeferenced by means of the GCPs coordinates or thanks to the onboard GNSS receiver. In a final step, the process allows obtaining high-resolution Digital Surface Models (DSMs).

Alternatively, LIDAR measurements give point clouds with really high density and precision, but photogrammetric UAV has lower costs [26] than laser scanners and surveys are less time-consuming [25]. Moreover, LIDAR surveys produce an enormous amount of data to be managed [1]. In particular, aerial LIDAR can be performed both using aircrafts

and drones, but the first case has no suitable costs for small areas, whereas drones with the sufficient payload to carry a laser scanner are still very expensive. Additionally, terrestrial laser surveys can be performed, but these are time consuming because of the needs of GNSS survey for georeferencing [29] and may produce point clouds with inhomogeneous density and precision.

Coastal studies usually rely on several parameters, such as the volume changes over time, the accumulating or eroding sectors, and the topographic profile changes. These can be evaluated by means of different elaborations of multitemporal surveys, depending on the employed technique [8–10,16,17,30,31].

This study aims to compare data from photogrammetric UAV and GNSS surveys over a common area located in Lido di Spina, a coastal stretch in the Emilia-Romagna region, Italy. Two multitemporal surveys were carried out in 2019 and 2020 using both techniques, a few days apart at most. GNSS surveys were performed using the RTK method, taking advantage of the presence of a benchmark belonging to the Coastal Geodetic Network (RGC) managed by Arpae [32]. The photogrammetric surveys were performed by means of a DJI Phantom 4 RTK, which is a low-cost instrument equipped with an onboard GNSS receiver recording phase observables to enable precise positioning. The analysis involved point-wise height differences between the two datasets, comparison of profile shapes along both cross-shore and along-shore sections, and volume variations in one year.

## 2. Study Area

The investigated area consists of a coastal stretch located in the municipality of Comacchio, in the Ferrara province. This place, which belongs to the National Reserve of Sacca del Bellocchio, is located south of the Lido di Spina town and north of the Gobbino channel (Figure 1). The site is part of the Regional Park of Delta del Po, and it is one of the most rural and biodiverse coastal areas of the Emilia-Romagna region, which has suffered less from anthropic impacts [33–36]. This area is also part of one of the cells identified by the Regional Agency for the Prevention, Environment, and Energy of Emilia-Romagna (Arpae), between Porto Corsini and Porto Garibaldi, where longshore transport is directed northwards [37]. The evolution of this cell is strongly dependent on the solid contribution of the Reno River, on the effects of Porto Corsini and Porto Garibaldi piers, on the solid transport direction, and, in a minor part, on the solid contribution of the Lamon river [36]. Since the first half of the 20th century, the progressive decrease in the sediment supply from the River Reno has been observed and, in conjunction with the divergent solid transport at the river mouth, has caused a serious erosion process which induced the mouth dismantling and a significant recession of adjacent beaches [35]. Different defence interventions have been realized to contrast the strong erosion process south of Lido di Spina, in the area of the Beach club Dolce Vita bathhouse (previously named Bagno Jamaica): five wood groynes (some of which strengthened with rocks), one rock groyne fixed to a grazing cliff, submerged barriers, and geotextile sandbags barriers. All these structures cannot properly protect the beach behind [37–40], therefore frequent nourishments are realized to restore the area using sand from the Logonovo channel. Despite the presence of defence structures and the realization of several beach nourishments over the years, the coastal stretch between the north shore of the Reno and Lido di Spina South still suffers from strong erosion.

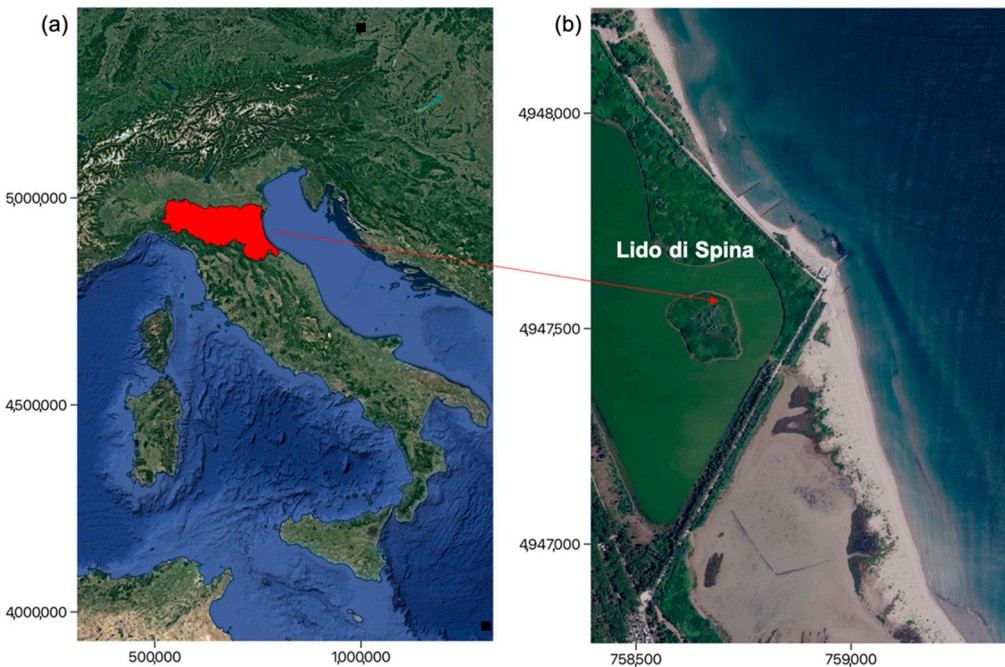

**Figure 1.** Lido di Spina beach (**b**) located between Porto Corsini and Porto Garibaldi, in the northern Adriatic coast of Emilia-Romagna (**a**). Base map from Agea orthophoto 2018 (https://geoportale.regione.emilia-romagna.it/servizi/servizi-ogc/elenco-capabilities-dei-servizi-wms#b, accessed on 20 July 2021) and Google satellite.

## 3. Materials and Methods

### 3.1. Surveys Data

The dataset of observations includes two multitemporal surveys (2019 and 2020) acquired both using GNSS and UAV photogrammetry (a few days apart at most). As these surveys were performed for different specific purposes (regular monitoring activities, monitoring of beach nourishment evolution, European projects) and by different operating companies, a potentially critical issue is related to the reference system. In this regard, a benchmark belonging to the Coastal Geodetic Network (PCPG0500) of ARPAE [7,32] was used to provide a proper alignment in the official national reference system ETRS89-ETRF2000 (2008.0). This fact can ensure internal consistency between data from different surveys allowing intercomparisons and proper DEM (Digital Elevation Model) differencing. All the analyses rely on the projected coordinates Northing and Easting, framed in the UTM32-WGS84-ETRS89-ETRF2000 system, while the geoid height of PCPG0500 was used to transform ellipsoidal heights to orthometric ones. The following paragraphs present in detail the technical specifications of the surveys.

#### 3.1.1. February 2019 Campaign

The GNSS survey realized in February 2019 involved an extended area of about 9.5 km, from Porto Garibaldi to the Reno River mouth. Cross and alongshore sections spaced 10.0–15.0 m were acquired in the emerged beach until the bathymetric of 0.7–0.8 m using a centimetre pole. Trimble R7 and Trimble R8s double-frequency GNSS receivers operating in RTK-OTF (On the Fly) mode were employed, placing the master station on the RGC benchmark. The average spatial distribution of acquired points along the sections was about 3.0–4.0 m and additional points were measured for all significative slope changes in the area. Horizontal and vertical accuracies related to the GNSS positioning in these conditions can be quantified as 2–3 cm and 3–5 cm, respectively.

The UAV survey involved the adjacent area of the Beach club Dolce Vita bathhouse, with an extension of about 1 km and a total area of about 135.900 m$^2$, only concerning the emerged beach. A DJI Phantom 4 RTK equipped with an onboard GNSS antenna was used.

Nevertheless, six targets were materialized in the area as GCPs and surveyed by means of geodetic class GNSS-RTK with the master station set on the PCPG0500 benchmark for the georeferencing of the photogrammetric model. The flight plan was designed to ensure an 80% overlap between the photograms in both directions.

### 3.1.2. January–February 2020 Campaign

The 2020 survey campaign was part of the European Project H2020 OPERANDUM (OPEn-air laboRAtories for Nature baseD solUtions to Manage environmental risk) (https://www.operandum-project.eu, accessed on 20 July 2021) for the study of Natural-Based Solutions as possible structures for the mitigation of extreme weather events. The purpose of this survey was to obtain data for a deep analysis of the selected area to set up the project of an artificial dune strengthened by a natural and biodegradable structure. For this reason, the surveyed area during this campaign was very limited compared to the previous one. The GNSS-RTK survey covered only the narrow area where the dune would be built, south to the Beach club Dolce Vita bathhouse, for a total of about 16.540 m$^2$. Data were acquired until the bathymetric of about 0.8 m, following profiles along sections spaced out of 7.0–10.0 m, with a spatial density of 1 point each 3.0–5.0 m. Concerning the technical characteristics of this GNSS survey, please refer to the previous section, since the same company operated using the same instruments.

The UAV survey of January 2020 involved a wider area compared to the GNSS measurements, about 155.470 m$^2$ of emerged beach until the bathymetric of 0.5 m. The same instruments and methodologies of the previous campaign in February 2019 were employed, with only one materialized GCP in the area. Note that in this case UAV survey was performed few days after the GNSS one.

Table 1 lists the main technical specifications of each UAV dataset.

**Table 1.** Summary of the key parameters of the available UAV photogrammetric surveys.

| Survey Data | 2019 | 2020 |
| :---: | :---: | :---: |
| Camera model | FC6310R | FC6310R |
| Resolution (px) | 5472 × 3648 | 5472 × 3648 |
| Focal length (mm) | 8 | 8 |
| Pixel size (μm) | 2.41 × 2.41 | 2.41 × 2.41 |
| N° frames | 563 | 638 |
| Flight altitude (m) | 104 | 108 |
| GSD (cm/px) | 2.87 | 2.64 |

### 3.2. Pre Processed Data

As for the UAV surveys, the elaboration processes which led to the point-clouds extraction were carried out by the operating companies. The photogrammetric models were generated using the software Agisoft Metashape v.1.7, which is based on SfM algorithms. The first step of the process was the alignment of different frames by identifying corresponding pixels. Here, the camera calibration parameters were estimated, also defining the camera location error estimates. GCPs coordinates were then imported into the software and identified on each frame to align the model in the chosen reference system The 3D point cloud was generated and densified during a second processing step. The model validation phase is a common practice for research studies, but reasonably it is not usually performed by companies engaged by third parties. In our case, we can consider the residuals obtained on the GCPs positions to evaluate the photogrammetric model accuracy. For the 2019 and 2020 surveys these values are 4.77 and 0.31 cm, respectively, for the horizontal component, whereas 4.10 and 2.67 cm in height.

### 3.3. Comparison Method

The height differences between the two data sources were calculated by using a processing tool included in the QGIS software package (http://www.qgis.org, accessed on 20 July 2020), "Point sampling tool", which is able to detect corresponding points from two datasets. This phase allowed to obtain a point-by-point comparison of height values on the GNSS points positions. With the adopted setup, negative differences result when photogrammetric UAV heights are higher than the GNSS ones, while positive values result from the opposite situation. The obtained height differences were stored as an attribute of the GNSS datasets, and for each survey two distinct maps were realized: in the first one all the points are represented and those associated with the highest discrepancies are emphasized using bright colours, while the second map represents only points characterized by height differences within 10 cm. The two maps have different associated colour scales. Considering the high spatial resolution of the photogrammetric UAV surveys, this approach allowed to directly compare independent GNSS points with the UAV corresponding ones, thus minimizing the influence of inherent interpolation errors on the GNSS data. The frequency histograms of height differences were computed, reporting their associated statistic parameters (mean values and standard deviations). The beach profiles, meaning graphs representing height variations with the increasing distance, were plotted along some of the GNSS-acquired sections (both along-shore and cross-shore).

The TIN Interpolation function of QGIS was used to calculate Digital Elevation Models. Spatial resolution, measurement accuracy, and interpolation algorithm are critical parameters of the DEM [17,30,41]. Therefore, the chosen spacing for the elaboration must take into account the real spatial resolution of acquired points to avoid possible mismatches and interpolation errors, especially with sparsely distributed data [17,26]. Since the spatial resolutions obtained from GNSS and UAV photogrammetry are usually different due to the inherent characteristics of each technique, different spacings for the DEMs were selected accordingly. Spacings values of 20 cm and 5 cm were chosen for the GNSS and UAVs data, respectively. Then, the one-year height variation map was calculated with both the techniques. To compare the two surveying methods also in terms of volume changes, these maps were limited to those areas common to the four surveys.

In the common approach for coastal studies, the comparison between DEMs of different epochs represents the evolution of the beach in terms of deposition or erosion. Height variation maps also allow calculating eroded/accumulated volumes, a key parameter in sandy-beach monitoring. Volume variations were computed using the "Raster surface volume" tool of QGIS, selecting the option to Subtract Volumes Below Base level (equal to 0) to consider both accumulated and eroded values.

## 4. Results

### 4.1. February 2019 Campaign

The survey campaign realized in 2019 allowed for comparison over an area of about 88.240 m$^2$. The spatial distribution of height differences over all the common points for the GNSS and UAV datasets is shown in Figure 2: Figure 2a shows all the values with a focus on higher residuals, while Figure 2b emphasizes only residual values up to 10 cm. Looking at the distribution, we can observe that most of the differences in Figure 2a are related to the $\pm$10 cm interval, with higher values located only in specific areas very close to the shore or where the data filtering was probably not sufficient. Figure 2b better characterizes the residual values within 10 cm, showing a quite homogeneous spatial distribution of the different ranges of values in the whole analysed area.

Obtained results are also reported in terms of mean values and standard deviations of the height differences, supported by the related frequency histograms. Due to possible tidal effects which can affect also surveys realized after a few hours, the nearshore zone is excluded from the histogram and the related statistical analysis. Figure 2 clearly highlights clusters of points having high residuals probably due to these effects.

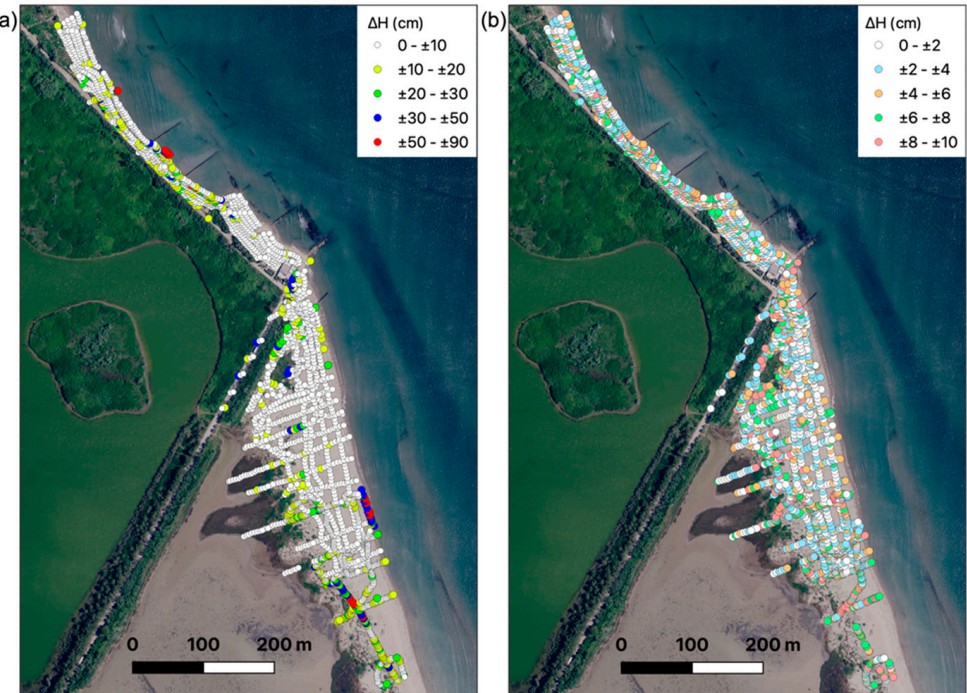

**Figure 2.** GNSS-acquired points and related height differences between RTK and photogrammetric UAV surveys for the 2019 campaign. (**a**) relates to all the values emphasizing the higher discrepancies; (**b**) relates to differences within 10 cm.

Figure 3 shows the histogram distribution of the height differences between the two datasets on 2165 points. This graph shows an almost symmetric distribution, with a mean value of −3 cm and an STD of 9 cm. Statistical data confirm an overall agreement between the two datasets, with 50% of the differences ranging between −6 cm and 1 cm and 90% of values ranging between −19 cm and 6 cm.

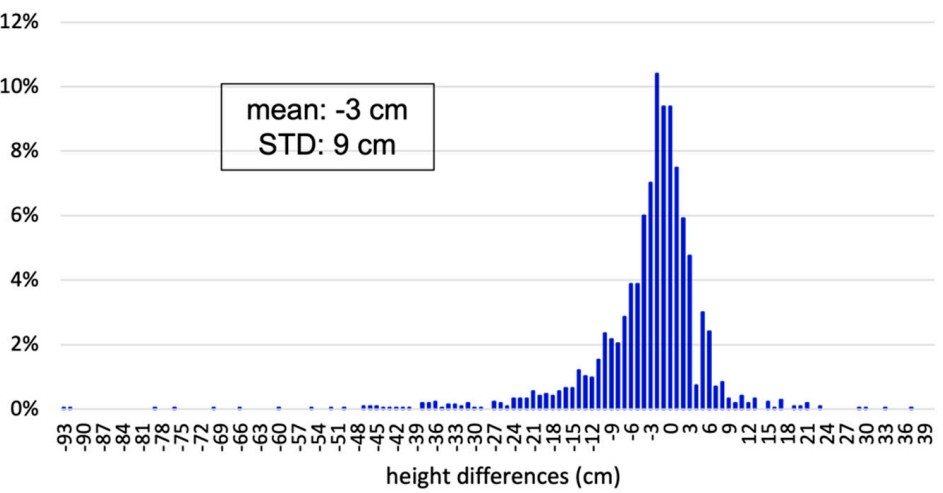

**Figure 3.** Histogram distribution of the height differences (GNSS-UAV) expressed in cm.

Four height profiles defined using both GNSS and UAV are shown in Figure 4. Three profiles (a, b, d) are related to cross-shore transects, and one (c) is related to the along-shore transect. Results are generally consistent at a few cm level and differences rising up to 30 cm are present in the cross-shore profile (a) where it goes through a bush area. The worst case is represented in Figure 4d, where a regular bias in the dm order seems to affect the GNSS and UAV points.

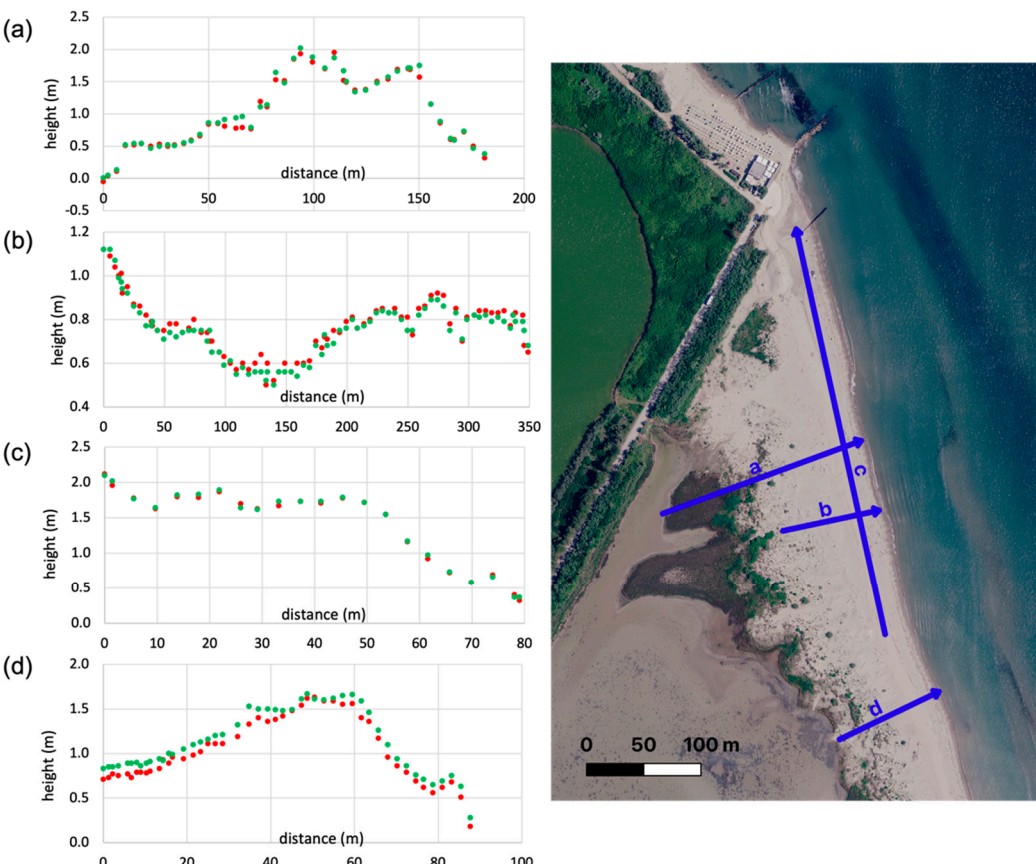

**Figure 4.** Examples of cross-shore (**a,b,d**) and along-shore (**c**) profiles of 2019 survey for GNSS (red) and UAV-derived (green) data.

### 4.2. January–February 2020 Campaign

Despite the limited spatial extension of the covered area, about 16.540 m², the GNSS survey of 2020 was characterized by a high density of points which allowed a proper point-wise comparison of height values. Figure 5a represents the height differences of all the common points, stressing the colour scale on higher values, while Figure 5b shows only the differences within 10 cm, sorted in different intervals. The distribution of higher values in Figure 5a is clearly concentrated near the shore, with just a few points located close to the vegetation. Residual values within the ±10 cm range (Figure 5b) show a quite homogeneous distribution on the whole area, without evident clusters of higher values.

Additionally, in this case, since higher differences are located near the shore (Figure 5a), points in this area have been excluded from the histogram and statistic computations. The histogram of height differences (Figure 6) shows the distribution for 556 points, where the mean is about 1 cm and the STD is 8 cm. The graph clearly shows a very limited number of high values, while 50% of the differences vary between −2 cm and 6 cm, and 90% are within the −8 cm and 10 cm range.

Figure 7 shows a comparison between cross-shore and along-shore profiles defined using the two techniques. Cross-shore profiles (a, b) show a general agreement between the two techniques. Only near the shore some differences up to 10 cm are present. In the along-shore profiles (c) located very close to the shore, a quite regular bias in the dm order can be found, while the same profile shape is described by both techniques. Figure 7d shows another cross-shore profile affected by some higher values in the dm order probably due to vegetated areas.

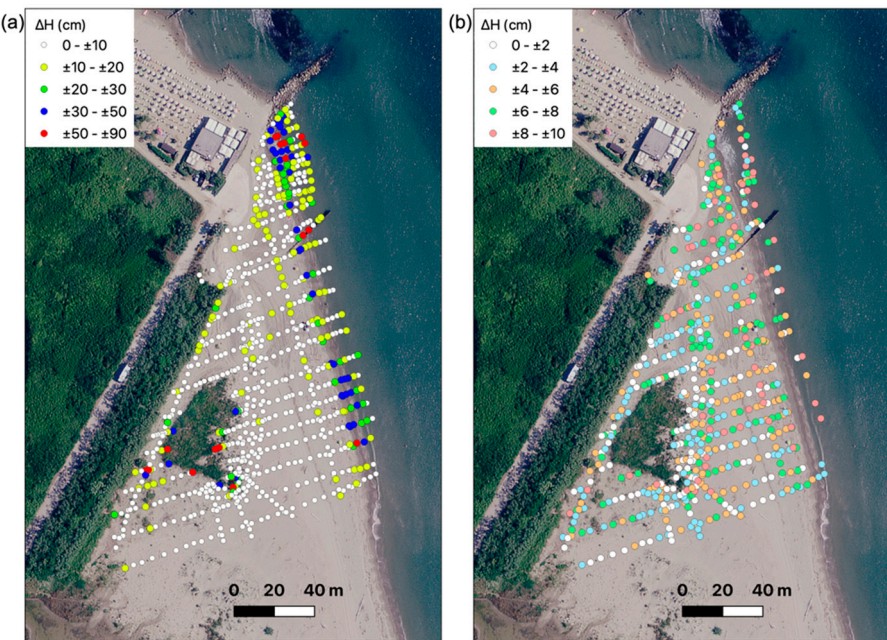

**Figure 5.** GNSS-acquired points and related height differences between RTK and photogrammetric UAV surveys for the 2020 campaign. (**a**) relates to all the values emphasizing the higher discrepancies; (**b**) relates to differences within 10 cm.

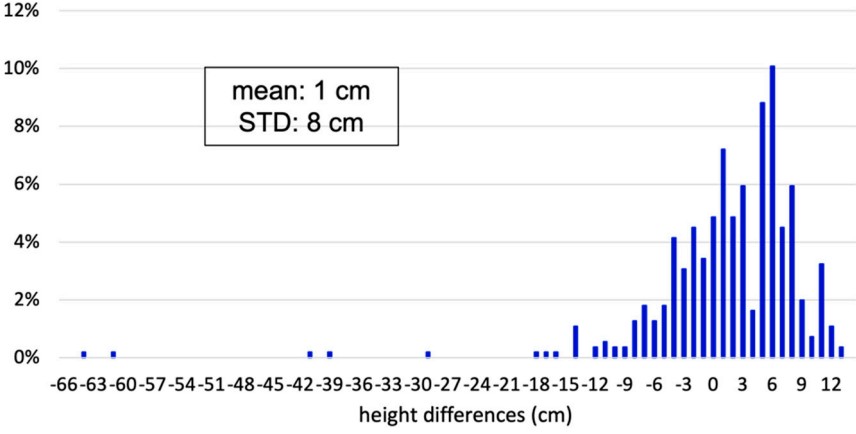

**Figure 6.** Histogram distribution of the height differences (GNSS-UAV) expressed in cm.

### 4.3. Volumes Monitoring

The comparison between 2020 and 2019 surveys using the same technique allowed obtaining the maps of height variation in case of monitoring using GNSS or UAV surveys only. Both these maps were computed over the common area (9.616 m$^2$) for all the involved surveys, excluding a central spot where the lack of points of the GNSS surveys could have affected the interpolation results. Additionally, for this computation, we excluded data in the nearshore area since affected by the delay between the 2020 surveys. Looking at Figure 8, regardless of the different spatial resolutions of the two datasets (a—GNSS; b—UAV) the overall sand variations are very similar. Both the one-year analysis denoted a sand accretion situation, with positive volumes of 2238 m$^3$ and 2233 m$^3$ for GNSS and UAV, respectively. The common approach for coastal studies exploits extended datasets composed of height variation maps, allowing to observe the differences experienced by the study area within a selected time span. Looking at Figure 8, a height increase between the two survey campaigns can be observed, with values from 30 cm up to 70 cm mostly concentrated along shore. Some local height differences of about 10–50 cm are localized near the vegetated area, and quite homogeneous values are distributed on the rest of the area.

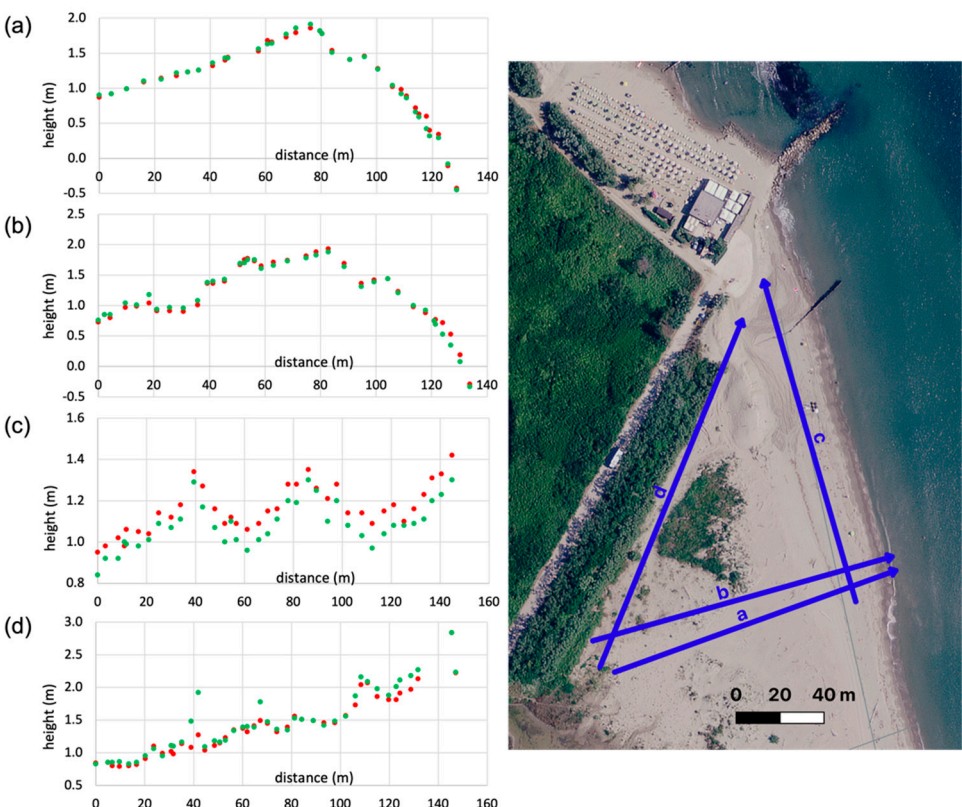

**Figure 7.** Examples of cross-shore (**a**,**b**,**d**) and along-shore (**c**) profiles of 2020 survey for GNSS (red) and UAV-derived (green) data.

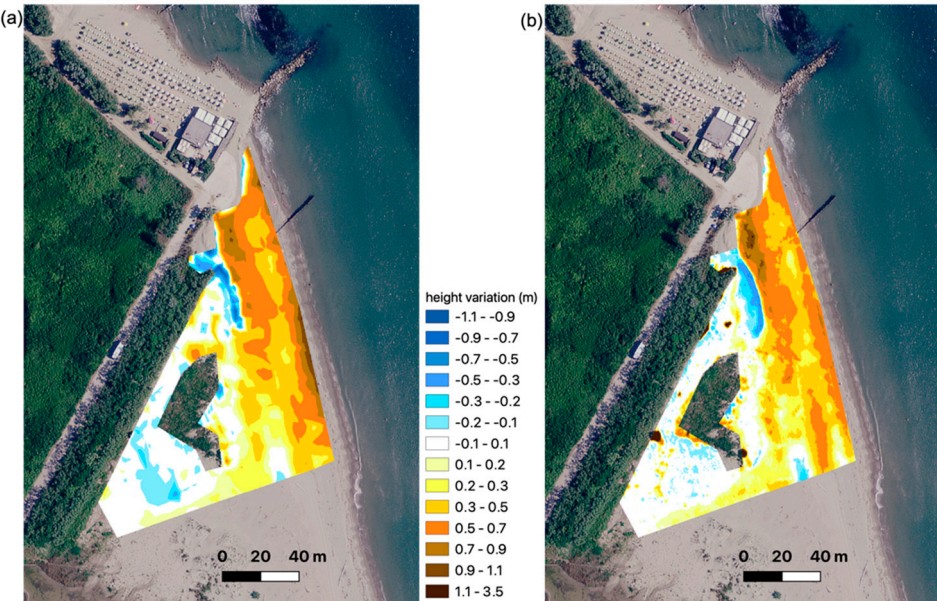

**Figure 8.** Maps of height variations between 2019 and 2020, for GNSS (**a**) and photogrammetric UAV surveys (**b**).

## 5. Discussion

The statistical distribution of the height differences computed for the 2019 and 2020 survey campaigns showed deviations of the means of about 3 cm and 1 cm, respectively. These biases can be due to (1) the stationing operations on the master station (RGC benchmark) or (2) managing of the pole during GNSS surveys, or (3) measurements over the

GCPs used to align UAV photogrammetric models. Regardless, the two techniques proved to be consistent and comparable: 50% of the height differences are lower than 7 and 8 cm (and 90% lower than 25 and 18 cm) for the 2019 and 2020 surveys, respectively. Looking at the spatial distribution of height differences, lower values are distributed quite uniformly in the whole area, while the higher variability is clearly located near the shore and in some vegetated areas. Intertidal zones or areas where saturated sand is present are known to be affected by a loss of accuracy in the SfM reconstruction [9]. Nevertheless, differences in the intertidal area are more evident in the comparison between the 2020 surveys, which were not conducted contemporarily but with few days of delay between the two techniques, therefore tides and waves may have changed the surface. Differently, vegetated areas can be critical when a comparison between GNSS and UAV datasets is performed. Photogrammetric methods inherently provide surface geometries (DSM) that cannot be directly compared to DTM, and data filtering is required to obtain ground-related data.

The STDs we found comparing GNSS and UAV over common points are 9–8 cm for the two surveys. Such a result is very similar to that found in [26], even though a higher-class UAV was used in that work at a much lower flying height. A similar comparison between GNSS surveyed profiles and UAV was performed by [11] using a low-cost drone flying about 53 m hight, which resulted in a 3 cm STD. This result is more consistent to what was found in [1] using a flying-wing drone, despite the poorer ground resolution. Using the same Phantom 4 RTK drones adopted here, Taddia et al. [24] found a STD of about 7.5 cm for nadiral UAV acquisition performed without GCPs. Such STD decreases to 3.4 cm when considering a sufficient number of GCPs. The differences between these results can be due to (1) the lower flying altitude (80 m instead of about 110 m) and (2) the reduced number and higher care paid in the GNSS survey of ad hoc control points instead of simpler GNSS profiles.

As for the beach profiles, the GNSS tracks along- and across-shore were considered; the height of each GNSS point was compared with the height of the closest point of the UAV-derived point cloud. Both the techniques allowed capturing profile variations with high precision since the data scattering is very small with respect to height variations along the profiles. The mean values of the biases along the profiles are at the cm level for all the analysed sections, except for the along-shore profiles (close to the shoreline) of the 2020 surveys, which were performed few days apart from each other.

After interpolating the GNSS-acquired points and the UAV-derived point-clouds for both the 2019 and 2020 campaigns, the DEM-to-DEM differences over time were computed, thus defining the volume changes estimated through the two different survey techniques. For this comparison, critical areas in the intertidal shoreline and vegetated areas were not considered. The sand monitoring led to very similar results using GNSS or UAV surveys alternatively, with volume differences of about 0.2%. Such a high coherence in volume changes definition can only be reached by paying attention to vegetated areas, which may strongly affect UAV monitoring if not properly managed.

## 6. Conclusions

The chosen site for this study lies along the Adriatic coast of the Emilia-Romagna region, in the National Reserve of Sacca del Bellocchio, which is particularly relevant from an environmental point of view. Thanks to the availability of two different survey campaigns realized concurrently with GNSS and photogrammetric UAV, this work aimed to compare these techniques in the context of coastal environment monitoring. The analysis focused on comparing the two surveying methods in terms of point-wise height differences, height profiles on cross-shore and along-shore sections, and volume changes over time.

In general, UAV photogrammetric surveys take advantage of the high spatial resolutions of the point clouds and allow describing also complex and uneven surfaces. Moreover, they allow surveying areas that are hardly accessible on the ground. Nevertheless, these surveys require a proper alignment to an established height reference, commonly obtained by means of GNSS surveys on a certain number of targets placed on the area, which

significantly increase the time spent on the field. Additionally, the post-processing of acquired images is a time-consuming phase for this kind of data. Using UAVs equipped with onboard RTK receivers, targets are potentially unnecessary and a good alignment can be obtained also with only one GCP [24], thus reducing the overall cost of the survey

Differently, GNSS surveys can be performed using real-time approaches and thus with no post-processing phases, obtaining coordinates directly aligned to the chosen reference. On the other hand, GNSS surveys can be very time-consuming if the terrain complexity requires a high density of the measured sections to avoid over-simplification of the surfaces in the interpolation phase.

Obtained results showed that the lower spatial resolution of the GNSS survey compared to the UAV photogrammetric one does not impact significantly on the estimated volumes if the terrain is quite regular or almost flat. This fact can be particularly relevant when Digital Elevation Maps are used as height input of Numerical Models applied in coastal management studies. Otherwise, the use of drone photogrammetry can provide a more accurate description of uneven terrains, without increasing the surveying effort as it should be done if using GNSS only.

Whatever the employed technique, when dealing with coastal analysis, a careful absolute georeferencing is a mandatory requirement to perform proper multitemporal comparisons.

**Author Contributions:** Conceptualization, L.T. and E.V.; methodology, L.T. and E.V.; formal analysis, S.G., L.T. and E.V.; resources, N.D.N.; writing—original draft preparation, L.T. and E.V.; writing—review and editing, L.T. and E.V.; supervision, S.G. All authors have read and agreed to the published version of the manuscript.

**Funding:** This research received no external funding.

**Conflicts of Interest:** The authors declare no conflict of interest.

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
