# Peer review of "GNSS and Photogrammetric UAV Derived Data for Coastal Monitoring: A Case of Study in Emilia-Romagna, Italy"

_jmse, doi:10.3390/jmse9111194_

Round 1

Reviewer 1 Report

Dear Editor and Authors, the reviewed manuscript was improved follwoing my suggestions and requirements. Overall, the main difect was the absence of UAV data processing, which has been added. Other minor modifications are required before publishing this work.

  • 5-9 please, follow guide to instruction: just add affiliation, the role/position of each author are not required and must not be added
  • Fig 1b, you can already mark the monitored study area 
  • Fig 2 shows points and area that are different from all others in the paper. SInce there are not any profiles compared, and further in the manuscript the northern part is not shown, in my opinon the survey of that area must be removed. In this way, histogram 3 and 6 would be comparable, now they show data from 2 different dataset. 
  • histrograms plot can be simplified i) choosing normal interval on xasix (starting with 0 > 0.2 0.4 0.6 , or intervals of 0.5 ecc.) and homogenize the plot of the two shown (now they have different xaxis and bins); ii) chosing a better data limit.
  • all over the paper, please homogenize decimal values of measures. Moreover, use m or cm when describing variation and/or differences, adopting always the same choise. In addition, avoid using "metres" and use always m. Finally, the text reports cm whereas it shows m in Figure (e.g. Fig.8), and viceversa. Please, homogenize. 
  • to me, resolution of 3D drone in micrometer does not make sence
  • in all Figures showing maps, please add axis and/or scale (better axis in m to give a better description of the study area)
  • in all Figures with graphs, please improve font size of axis and legend, numbers are not well readable
  • section 4.4, showing morphological changes without a proper description of waves and tide over the period (or at least prior the surveys, e.g. storms) is quite superficial. Please, i) rectify the term "evolution", these are just two "snapshots"/"surveys" in two different dates and does not show any evolution; ii) rectify the term "nearshore patch". The nearshore domain may have different definition and area limits, however you just surveyed subaerial beach, or foreshore, or you meant intertidal area (although in a  microtidal beach this area is quite limited) iii) recitfy the term "prevalent"
  • 416-418 "between two acquired dataset", please rectify. Drone acquires images that do not have any spatial info (they just have some metadata) before applying SFM, GNSS acquires already measures.
  • 420-432 confusing and not well written. Why not simply saying and resuming: UAV allows faster survey, but requires experts for data processing (you asked a company to survey that reason). GNSS survey is easier and can be done by briefly trained personell, but it requires mush more effort in the field. 
  • 433-442 please, avoid to generalize in Conclusions and just resume your numerical results. Differences were due to some vegetation and saturated sand, indicate and resume numerical results from histograms 
  • "lower spatial resolution of the GNSS survey does not impact significantly on the estimated volumes if the terrain is quite regular or close to being flat": this is a non-sense. It is obvious that lower the magnitude/volume, lower will be differences (for this reason, for many survey techniques comparison, it is used the normalized error).

Author Response

RC#1

5-9 please, follow guide to instruction: just add affiliation, the role/position of each author are not required and must not be added

AA#1

Dear Reviewer, you are right. Thank you.

RC#2

Fig 2 shows points and area that are different from all others in the paper. SInce there are not any profiles compared, and further in the manuscript the northern part is not shown, in my opinon the survey of that area must be removed. In this way, histogram 3 and 6 would be comparable, now they show data from 2 different dataset.

AA#2

Dear Reviewer, the differences between the acquired data during the 2019 and 2020 surveys are explained in the Material and Methods section. We aimed to compare all the available data for both cases, and this is the reason why Figure 2 shows a wider area with respect to Figure 6.

RC#3

histrograms plot can be simplified i) choosing normal interval on xasix (starting with 0 > 0.2 0.4 0.6 , or intervals of 0.5 ecc.) and homogenize the plot of the two shown (now they have different xaxis and bins); ii) chosing a better data limit.

AA#3

Dear Reviewer, we changed the intervals on the x-axis for both Figure 3 and Figure 6.

RC#4

all over the paper, please homogenize decimal values of measures. Moreover, use m or cm when describing variation and/or differences, adopting always the same choise. In addition, avoid using "metres" and use always m. Finally, the text reports cm whereas it shows m in Figure (e.g. Fig.8), and viceversa. Please, homogenize.

AA#4

Dear Reviewer, we decided to use cm to describe variations and differences. We also modified the Figures legends according to this. However, as in Figure 8 values are of a greater order, we maintained the legend in m.

RC#5

to me, resolution of 3D drone in micrometer does not make sence

AA#5

Dear Reviewer, we didn’t use micrometers in the manuscript, except for the dimension of the pixels.

RC#6

in all Figures showing maps, please add axis and/or scale (better axis in m to give a better description of the study area)

AA#6

Dear Reviewer, we added the scale in all the Figures with maps, as the axis were present in Figure 1.

RC#7

In all Figures with graphs, please improve font size of axis and legend, numbers are not well readable

AA#7

Dear Reviewer, you are right. We increased the font size in the new Figures.

RC#8

section 4.4, showing morphological changes without a proper description of waves and tide over the period (or at least prior the surveys, e.g. storms) is quite superficial. Please, i) rectify the term "evolution", these are just two "snapshots"/"surveys" in two different dates and does not show any evolution; ii) rectify the term "nearshore patch". The nearshore domain may have different definition and area limits, however you just surveyed subaerial beach, or foreshore, or you meant intertidal area (although in a  microtidal beach this area is quite limited) iii) recitfy the term "prevalent"

AA#8

Dear Reviewer, you are right. Thi short paragraph aimed to underline that by using this method of comparison with a more extended dataset is possible to monitor coastal changes over long periods. We changed these sentences according to your suggestions.

RC#9

416-418 "between two acquired dataset", please rectify. Drone acquires images that do not have any spatial info (they just have some metadata) before applying SFM, GNSS acquires already measures.

AA#9

Dear Reviewer, you are right. The datasets obtained by GNSS and UAV photogrammetry are not directly comparable. We replaced “comparing the two acquired datasets” with “comparing the two data sources”.

RC#10

420-432 confusing and not well written. Why not simply saying and resuming: UAV allows faster survey, but requires experts for data processing (you asked a company to survey that reason). GNSS survey is easier and can be done by briefly trained personell, but it requires mush more effort in the field. 

AA#10

Dear Reviewer, we did not refer only to the complexity or simplicity of the survey or to the differences in time needed for the field operations and the post-processing. We pointed out also other aspects, as the need for an additional ground survey for a proper alignment, the difference in field operations while performing a GNSS survey depending on the survey requirements in terms of spatial resolution.

We do not believe that this comparison could be replaced with these two sentences.

RC#11

433-442 please, avoid to generalize in Conclusions and just resume your numerical results. Differences were due to some vegetation and saturated sand, indicate and resume numerical results from histograms 

AA#11

Dear Reviewer, all this information are in the Discussion section. We believe that the Conclusion should not include numerical data, while more general comments on the research and possible applications.

RC#12

"lower spatial resolution of the GNSS survey does not impact significantly on the estimated volumes if the terrain is quite regular or close to being flat": this is a non-sense. It is obvious that lower the magnitude/volume, lower will be differences (for this reason, for many survey techniques comparison, it is used the normalized error).

AA#12

Dear Reviewer, it is known that computed volumes depend on the spatial resolution of the points and the terrain irregularities. With this sentence, we did not refer to differences in the accumulated/eroded volumes for a single map of height variation produced with one selected technique. We were pointing out that in these cases (“if the terrain is quite regular or close to being flat”) it is possible to obtain a proper evaluation in terms of volume also using a quite low spatial resolution (as one of our GNSS surveys).

Reviewer 2 Report

This paper focuses on the coastal monitoring by using GNSS and photogrammetric UAV data. Two couple of surveys by using the two methods over a common area have been compared, which seems meaningful for this topic. I recommend this paper for publication and I just have some advices as follows.

  1. “GNSS surveys are particularly suitable for the monitoring of coastal areas thanks to the absence of obstacles that could reduce the sky visibility”, the expressions are not totally right. The multipath effects (Zhang Z, Yuan H, Li B, He X, Gao S. Feasibility of easy-to-implement methods to analyze systematic errors of multipath, differential code bias, and inter-system bias for low-cost receivers. GPS Solutions, 2021, 25: 116.) can be significant at this time when the receiver is close to the sea areas. The authors should address this topic in the revision.
  2. What are the X-axis and Y-axis stand for? And what are their units, please supplement;
  3. It is advised that the authors should add a table like Table 1 to explain the processing strategies of GNSS data of the two surveys in Section 3;
  4. Figures 3 and 6 should be drawn more beautiful, especially for Figure 6.

Author Response

RC#1

GNSS surveys are particularly suitable for the monitoring of coastal areas thanks to the absence of obstacles that could reduce the sky visibility”, the expressions are not totally right. The multipath effects (Zhang Z, Yuan H, Li B, He X, Gao S. Feasibility of easy-to-implement methods to analyze systematic errors of multipath, differential code bias, and inter-system bias for low-cost receivers. GPS Solutions, 2021, 25: 116.) can be significant at this time when the receiver is close to the sea areas. The authors should address this topic in the revision.

AA#1

Dear Reviewer, we did not find any feedback about the impact of water on the multipath on this reference. Moreover, in our case the quality control on the master station acquisition did not highlight significant effects and geodetic class receivers have been employed, so we believe that this issue does not arise for our measures.

RC#2

What are the X-axis and Y-axis stand for? And what are their units, please supplement;

AA#2

Dear Reviewer, we did not find any references to “X-axis” and “Y-axis” in our manuscript.

RC#3

It is advised that the authors should add a table like Table 1 to explain the processing strategies of GNSS data of the two surveys in Section 3.

AA#3

Dear Reviewer, in the text we mentioned the receiver models and the employed acquisition method. We don’t have more specific data obtained from the surveying companies, but we do not think that other parameters could significantly impact the results (cut-off angle, RTK protocol,…). For these reasons, we believe that a Table for the GNSS datasets would be useless. 

RC#4

Figures 3 and 6 should be drawn more beautiful, especially for Figure 6.

AA#4

Dear Reviewer, we increased the font size and homogenized the axis intervals for Figure 3 and Figure 6.

Reviewer 3 Report

According to the current appearance of the manuscript, it may be the second round of review. As  not seeing the comments raised in the first round of review, I could not judge whether the modification successfully addressed those comments. After careful reading, I believe this manuscript has almost reached the publishing standard, so it is recommended to be published in Journal of Marine Science and Engineering. However, there are still several minor modifications that should be made.

1. Line 186, change the "overlay " into "overlapping".

2. In Figure 6. the extra rectangle around the mean and STD should be removed. 

Author Response

Dear Reviewer, thank you for your suggestions. We accepted both the corrections.